# Population-level viremia predicts HIV incidence at the community level across the Universal Testing and Treatment Trials in eastern and southern Africa

Joseph Larmarange[1]*, Pamela Bachanas[2], Timothy Skalland[3], Laura B. Balzer[4], Collins Iwuji[5], Sian Floyd[6], Lisa A. Mills[7], Deenan Pillay[8], Diane Havlir[9], Moses R. Kamya[10], Helen Ayles[11,12], Kathleen Wirth[13], François Dabis[14], Richard Hayes[6], Maya Petersen[4], for the UT³C consortium¶

1 Centre Population et Développement, Université Paris Cité, IRD, Inserm, Paris, France, 2 Division of Global HIV/AIDS and TB, Centers for Disease Control and Prevention, Atlanta, GA, United States of America, 3 Fred Hutchinson Cancer Center, Seattle, WA, United States of America, 4 Division of Biostatistics, School of Public Health, University of California, Berkeley, California, United States of America, 5 Department of Global Health and Infection, Brighton and Sussex Medical School, University of Sussex, Falmer, United Kingdom, 6 Department of Infectious Disease Epidemiology, London School of Hygiene and Tropical Medicine, London, United Kingdom, 7 Division of Global HIV and TB, Centers for Disease Control and Prevention, Gaborone, Botswana, 8 Division of Infection & Immunity, University College London, London, United Kingdom, 9 Department of Medicine, University of California San Francisco, San Francisco, CA, United States of America, 10 Department of Medicine, Makerere University Kampala, Uganda and the Infectious Diseases Research Collaboration, Kampala, Uganda, 11 Clinical Research Department London School of Hygiene & Tropical Medicine, London, United Kingdom, 12 Zambart, University of Zambia School of Public Health, Lusaka, Zambia, 13 Harvard T.H. Chan School of Public Health, Boston, MA, United States of America, 14 Université Bordeaux, ISPED, Centre INSERM U1219-Bordeaux Population Health, Bordeaux, France

¶ Composition of the UT³C consortium is provided in the Acknowledgments
* joseph.larmarange@ird.fr

**Data Availability Statement:** A dedicated dataset and an R script are provided for replication (file S2 in supplementary materials).

## Abstract

Universal HIV testing and treatment (UTT) strategies aim to optimize population-level benefits of antiretroviral treatment. Between 2012 and 2018, four large community randomized trials were conducted in eastern and southern Africa. While their results were broadly consistent showing decreased population-level viremia reduces HIV incidence, it remains unclear how much HIV incidence can be reduced by increasing suppression among people living with HIV (PLHIV). We conducted a pooled analysis across the four UTT trials. Leveraging data from 105 communities in five countries, we evaluated the linear relationship between i) population-level viremia (prevalence of non-suppression–defined as plasma HIV RNA >500 or >400 copies/mL–among all adults, irrespective of HIV status) and HIV incidence; and ii) prevalence of non-suppression among PLHIV and HIV incidence, using parametric g-computation. HIV prevalence, measured in 257 929 persons, varied from 2 to 41% across the communities; prevalence of non-suppression among PLHIV, measured in 31 377 persons, from 3 to 70%; population-level viremia, derived from HIV prevalence and non-suppression, from < 1% to 25%; and HIV incidence, measured over 345 844 person-years (PY), from 0.03/100PY to 3.46/100PY. Decreases in population-level viremia were

**Funding:** The HPTN 071 (PopART) trial was supported by the National Institute of Allergy and Infectious Diseases (NIAID) under Cooperative Agreements UM1-AI068619, UM1-AI068617, and UM1-AI068613, with funding from the U.S. President's Emergency Plan for AIDS Relief (PEPFAR); the International Initiative for Impact Evaluation with support from the Bill and Melinda Gates Foundation; as well as by NIAID, the National Institute on Drug Abuse (NIDA), and the National Institute of Mental Health (NIMH), all part of the National Institutes of Health NIH. RH and SF received funding from the UK Medical Research Council (MRC) and the UK Foreign, Commonwealth and Development Office (FCDO) under the MRC/FCDO Concordat agreement and is also part of the EDCTP2 programme supported by the European Union. Grant Ref: MR/R010161/1. The SEARCH trial was supported by the Division of AIDS, National Institute of Allergy and Infectious Diseases of the National Institutes of Health (awards U01AI099959, UM1AI068636, and R01 AI074345-06A1); the President's Emergency Plan for AIDS Relief; and Gilead Sciences, which provided tenofovir–emtricitabine (Truvada) in kind. The ANRS 12249 TasP trial was sponsored by the French National Agency for AIDS and Viral Hepatitis Research (ANRS; grant number, 2011-375), and funded by the ANRS, the Deutsche Gesellschaft für Internationale Zusammenarbeit (GIZ; grant number, 81151938), and the Bill & Melinda Gates Foundation through the 3ie Initiative. The Ya Tsie trial was supported by the President's Emergency Plan for AIDS Relief through the Centers for Disease Control and Prevention (CDC) (cooperative agreements U01 GH000447 and U2G GH001911); the National Institutes of Health; the Oak Foundation; and the Sub-Saharan African Network for TB/HIV Research Excellence (SANTHE), a Developing Excellence in Leadership, Training, and Science (DELTAS) Africa initiative (grant DEL-15-006, through Wellcome Trust 107752/Z/15/Z). The funders had no role in study design, data collection and analysis, decision to publish, or preparation of the manuscript.

**Competing interests:** I have read the journal's policy and the authors of this manuscript have the following competing interests: Dr. Collins Iwuji has received research grants and honorarium for consulting services from Gilead Sciences. Dr. Diane V. Havlir reports receiving non-financial support from Gilead Sciences. All other authors declare no competing interests.

strongly associated with decreased HIV incidence in all trials (between 0.45/100PY and 1.88/100PY decline in HIV incidence per 10 percentage points decline in viremia). Decreases in non-suppression among PLHIV were also associated with decreased HIV incidence in all trials (between 0.06/100PY and 0.17/100PY decline in HIV incidence per 10 percentage points decline in non-suppression). Our results support both the utility of population-level viremia as a predictor of incidence, and thus a tool for targeting prevention interventions, and the ability of UTT approaches to reduce HIV incidence by increasing viral suppression. Implementation of universal HIV testing approaches, coupled with interventions to leverage linkage to treatment, adapted to local contexts, can reduce HIV acquisition at population level.

## Introduction

Early initiation of antiretroviral treatment (ART), offers both individual and population-level benefits, in terms of reductions in morbidity and mortality [1,2] and decrease of HIV sexual transmission [3]. Since 2004, the rapid scale-up of antiretroviral therapy in sub-Saharan Africa [4] has resulted in substantial population-level reductions in HIV-related mortality [4,5]. There is also population-level evidence that ART scale-up has reduced new HIV infections, including observational data from rural KwaZulu-Natal, South Africa, demonstrating a strong inverse association between ART coverage and HIV incidence [4,6]. The Joint United Nations Programme on HIV/AIDS (UNAIDS) has fixed ambitious 95-95-95 targets for 2025, i.e. that 95% of people living with HIV (PLHIV) know their HIV status, that among them 95% are on ART, of whom 95% are virally suppressed [7].

Universal testing and treatment (UTT) strategies aim to optimize the population-level benefits of ART through (i) regular HIV testing of all adult members of a community; (ii) the offer of immediate ART initiation to all individuals diagnosed HIV-positive, regardless of immunological or clinical staging; and (iii) supported linkage to care and ART delivery. Between 2012 and 2018, four large community randomized trials were conducted in eastern and southern Africa to evaluate the impact of UTT strategies on HIV incidence and other outcomes: the HPTN 071 Population Effects of Antiretroviral Therapy to Reduce HIV Transmission (PopART) trial [8] in Zambia and South Africa (Western Cape), the Sustainable East Africa Research in Community Health (SEARCH) trial [9] in Kenya and Uganda, the ANRS 12249 Treatment as Prevention (TasP) trial [10] in South Africa (KwaZulu-Natal), and the Botswana Combination Prevention Project (BCPP) Ya Tsie trial [11] in Botswana.

While the UTT trial results were broadly consistent with the hypothesis that decreased population-level viremia reduces HIV incidence [12], how much HIV incidence can be reduced at a population level by increasing suppression among PLHIV remains an open question. While the answer will likely vary depending on factors such as mobility, sexual network structure, and risk behaviours, quantifying this relationship is essential for both projecting future trends in incidence and understanding likely incidence impacts from investments in testing and treatment strategies.

To address this question, we conducted a pooled analysis across the four UTT trials, leveraging the fact that they were conducted across a wide range of settings in eastern and southern Africa, and all conducted rigorous longitudinal population-level assessments of HIV RNA measures, HIV prevalence, and HIV incidence. We aimed to describe the direction and strength of the relationship between population-level viremia (the proportion of an entire

community, irrespective of HIV status, with non-suppressed plasma HIV RNA) and HIV incidence and to assess how this relationship changed when performing a cross-gendered analysis (assuming that transmission is mainly through heterosexual sex), by looking at the relationship between men's population-level viremia and women's HIV incidence, and between women's population-level viremia and men's HIV incidence. Further, because (i) the relationship between population viremia and incidence is likely to be confounded by factors affecting both HIV prevalence and incidence; and (ii) population-level interventions can modify non-suppression among PLHIV more directly than population viremia, we evaluated the association between prevalence of non-suppression among PLHIV and HIV incidence and, in each trial, estimated the expected incidence reduction during the trial associated with the observed increase in suppression among PLHIV.

## Methods

### Ethics statement

This cross-trial analysis is a secondary analysis of data that have already been individually published. Ethics approval was granted for the respective four trials from the relevant ethics committees. Consent procedures are detailed in the primary trial publications.

Ethical approval for the PopART trial was granted by ethics committees of the London School of Hygiene & Tropical Medicine, the University of Zambia and Stellenbosch University. The trial was registered on ClinicalTrials.gov: NCT01900977.

The SEARCH trial protocol was approved by the ethics committees at the University of California, San Francisco; the Kenya Medical Research Institute; and the Makerere University School of Medicine in Uganda; and registered on ClinicalTrials.gov: NCT01864603.

The ANRS 12249 TasP trial was approved by the Biomedical Research Ethics Committee of the University of KwaZulu-Natal, South Africa (BFC 104/11) and the Medicines Control Council of South Africa; and registered on ClinicalTrials.gov: NCT01509508 and in the South African National Clinical Trials Register: DOH-27-0512-3974.

The Ya Tsie trial was approved by institutional review boards at the US Centers for Disease Control and Prevention and the Botswana Ministry of Health and Wellness; and registered on ClinicalTrials.gov: NCT01965470.

### Study settings

Each trial has published its own protocol [8–11] as well as its primary outcome results [13–16]. In total, the trials enrolled 105 communities (clusters) and delivered interventions reaching almost 1.5 million people [17]. The trials implemented different interventions (Table 1), data collection tools and methods (Table 2).

All trials included a comprehensive set of interventions to provide universal access to HIV testing and facilitate linkage to HIV care [17]. PopART, TasP, and Ya Tsie implemented door-to-door home-based services provided by community health workers. SEARCH used a hybrid model of multi-disease community-based health fairs and mobile outreach. Whilst PopART and Ya Tsie implemented **universal testing** only in their intervention arm, such interventions were also implemented in the control arm for SEARCH and TasP at baseline, and repeated every six months for TasP.

All trials offered a wide range of additional services to support rapid ART initiation (**universal treatment**) regardless of CD4 count or clinical staging in their intervention arms. ART was offered according to respective national guidelines in the control arms (and in arm B for PopART). From 2016, and following new WHO guidelines [18], PopART, SEARCH, and Ya Tsie implemented universal treatment in all arms, whilst the TasP trial had already completed

**Table 1. Trial key features, HIV prevalence at baseline, prevalence of non-suppression among PLHIV at baseline, and observed overall HIV incidence, by trial and arm.**

| Trial | PopART | | | SEARCH | | TasP | | Ya Tsie | |
|---|---|---|---|---|---|---|---|---|---|
| Country | South Africa & Zambia | | | Kenya & Uganda | | South Africa | | Botswana | |
| Timeline | 2013–2018 | | | 2013–2017 | | 2012–2016 | | 2013–2018 | |
| Arm | C | I (arm A) | I (arm B) | C | I | C | I | C | I |
| Universal testing | - | ü | ü | ü | ü | ü | ü | - | ü |
| • approaches | | Door-to-door, mobile outreach | Door-to-door, mobile outreach | Multi-disease campaigns, door-to-door | Multi-disease campaigns, door-to-door | Door-to-door, mobile clinics (last round) | Door-to-door, mobile clinics (last round) | | Door-to-door, mobile clinics |
| • testing frequency | | Annual | Annual | Baseline | Annual | 6 monthly | 6 monthly | | Baseline, ongoing targeted |
| Universal treatment | | | | | | | | | |
| • from baseline | - | ü | - | - | ü | - | ü | - | ü |
| • from 2016 | ü | ü | ü | ü | ü | trial closure during first 2016 semester | | ü | ü |
| HIV prevalence at baseline [95% CI] | 21.1% [17.9 to 24.2] | 20.1% [16.9 to 23.2] | 19.6% [16.5 to 22.8] | 10.1% [6.4 to 13.7] | 10.4% [6.7 to 14.1] | 30.8% [27.4 to 34.5] | 29.3% [25.3 to 33.5] | 26.5% [19.1 to 33.9] | 26.9% [19.5 to 34.3] |
| Prevalence of non-suppression among PLHIV at baseline [95% CI] | 48.9% [43.8 to 54.0] | 45.5% [40.4 to 50.6] | 44.8% [39.7 to 49.9] | 58.5% [56.3 to 60.6] | 58.1% [53.9 to 62.2] | 74.0% [71.0 to 76.7] | 76.5% [74.1 to 78.8] | 28.4% [26.2 to 30.6] | 29.8% [27.7 to 32.0] |
| Overall HIV incidence | | | | | | | | | |
| • per 100 person-years | 1.55 | 1.45 | 1.06 | 0.27 | 0.25 | 2.27 | 2.11 | 0.92 | 0.59 |
| • reduction (I vs C) | A vs C: not significant B vs C: 30% reduction | | | not significant | | not significant | | 31% reduction | |

C: Control–I: Intervention–py: Persons-years–CI: Confidence intervals. Sources for HIV incidence: [13–16]. HIV prevalence and prevalence of non-suppression at baseline were adjusted as described in Table 2.

follow-up and transferred patients to the public ART programme. Other interventions, such as prevention services, were implemented and are summarised elsewhere [17].

## Measures

To enumerate the population, SEARCH and TasP conducted population-wide household census. TasP updated residency data every six months at each intervention round, whilst SEARCH conducted a census prior to intervention delivery at baseline and endline. PopART randomly selected ≈2 000 adults per community at baseline to constitute a population cohort surveyed annually, and enrolled new individuals at months 12 and 24. In Ya Tsie, a 20% random sample of household-like structures identified by satellite imagery was selected for enumeration, enrolment, and constitution of a population cohort.

  **HIV prevalence** and the **prevalence of non-suppression among PLHIV** were estimated at a community level for each of the 105 communities enrolled in the four trials. Due to differences in data availability, **HIV prevalence** was estimated at baseline for PopART and Ya Tsie, and at or around midpoint for TasP and SEARCH (Tables 2 and S1). Viral suppression was defined as plasma HIV RNA level <400 copies/mL in PopART, TasP, and Ya Tsie, and <500 copies/mL in SEARCH. The **prevalence of non-suppression** among adult PLHIV was estimated at baseline, around midpoint, and at endline in each trial (Tables 2 and S1). **Population-level viremia**, also known as the prevalence of detectable virus and defined as the

**Table 2. Data sources for the current analysis and estimation approaches for community-level HIV prevalence, non-suppression among PLHIV, and HIV incidence, by trial.**

| Trial | PopART | SEARCH | TasP | Ya Tsie |
|---|---|---|---|---|
| **Number of communities** (arms x communities / arm) | 21 (3 × 7) | 32 (2 × 16) | 22 (2 × 11) | 30 (2 × 15) |
| **Communities per country** | South Africa: 9 (3 × 3) Zambia: 12 (3 × 4) | Kenya: 12 (2 × 6) Uganda: 20 (2 × 10) | South Africa: 22 (2 × 11) | Botswana: 30 (2 × 15) |
| **Eligibility criteria for inclusion in current analysis** | | | | |
| • Age | 18–44 years (population cohort) | ≥15 years | ≥16 years | 16–64 years |
| • Residency | Participant defined residence in a household in the trial community | Participant defined residence in a household in the trial community | Resident for ≥4 nights per week within the household in general | On average ≥3 nights per month and more nights in the household than any other in the community over the preceding 12 months |
| • Nationality | No restriction based on nationality | No restriction based on nationality | No restriction based on nationality | Documented citizenship of Botswana or marriage to a citizen, due to national treatment guidelines |
| **HIV prevalence** | | | | |
| • Timing | Baseline | Average of all available measures (baseline and endline for control arm, annually for intervention arm) | Midpoint | Baseline |
| • Population | Representative sample of ≈2 000 adults per community constituting a survey population cohort | Open cohort of the entire population, updated through yearly campaigns | Open cohort of the entire population, updated through every six months home-based visits | Representative sample of ≈20% of households within the communities constituting a survey population cohort |
| • Adjustment | Age-sex standardization | Adjustment on age, sex, demographics, and prior testing | Partially adjusted (imputation for those with at least one HIV status observed at any given point) | Age-sex standardization to the 2011 Botswana Population Census |
| **Non-suppression among PLHIV at midpoint** | | | | |
| • Timing | M24 | Average of all available measures (baseline and endline for control arm, annually for intervention arm) | Midpoint | Midpoint (year 2) |
| • Population | Individuals HIV+ from the survey population cohort (including seroconverters, and individuals newly enrolled at M12 & M24) | Individuals HIV+ from the open cohort of the entire population (including seroconverters and immigrants) | Individuals HIV+ from the open cohort of the entire population (including seroconverters and immigrants) | Individuals HIV+ from the survey population cohort (closed cohort excluding immigrants, excluding seroconverters) |
| • Definition of non-suppression | >400 copies/mL | >500 copies/mL | >400 copies/mL | >400 copies/mL |
| • Adjustment | Age-sex standardization | Estimated for all PLHIV, including those undiagnosed. Adjustment on age, sex, demographics, prior testing, prior treatment, and prior suppression | Interpolation to decide if suppressed at a given time Considered as non-suppressed if not documented (including private sector) or if not into care | Age-sex standardization |
| **HIV incidence** | | | | |

*(Continued)*

**Table 2.** (Continued)

| Trial | PopART | SEARCH | TasP | Ya Tsie |
|---|---|---|---|---|
| • Timing | Between months 12 and 36 | Between months 0 and 36 | Between months 0 and 18–40 (communities had different follow-up times) | Between months 0 and 30 (communities had different follow-up times) |
| • Population | Individuals 18–44 and HIV- at baseline from the survey population cohort (open incidence cohort, including individuals newly enrolled at M12 & M24) | Individuals HIV- at baseline (closed incidence cohort, excluding immigrants) and still resident at endline (excluding outmigrants) | Open cohort (including immigrants) with varying individual follow-up time (from first to last known HIV status), estimated date of seroconversion, taking into account person-time at risk within the trial area and excluding seroconversions who occurred when not resident | Individuals HIV- at baseline from the survey population cohort (closed incidence cohort, excluding immigrants) |
| • Adjustment | Age-sex standardization | None (done in sensitivity analysis) | None | None |

prevalence of non-suppressed HIV infections among all resident adults, irrespective of HIV status, was obtained by multiplying the prevalence of non-suppression (among PLHIV) at midpoint and HIV prevalence.

**HIV incidence** (per 100 person years) in each community was estimated using repeat HIV testing to identify seroconversions, in either a closed incidence cohort (SEARCH and Ya Tsie) or an open population cohort (PopART and TasP). Incidence rate was calculated for the risk period ranging from either trial start (BCPP, SEARCH, and TasP), or from 12 months after trial start (PopART) until trial closure (Tables 2 and S1).

## Analysis

The relationship between population-level viremia and HIV incidence was estimated, using data aggregated at the community level from both intervention and control arms, through a linear regression allowing for trial-specific slopes and intercepts to account for variations in the trials' context, design, and measures. The unit of analysis was trial cluster: each cluster contributed to one observation point, with one measure of viremia and one measure of incidence. The clusters were unweighted in the analysis. As a sensitivity analysis, we also explored a quadratic relationship; as it complicated interpretation without improving fit, results are not reported here. Additional sensitivity analyses included incorporating: 1) trial-arm specific slopes and intercepts; and, 2) trial and country specific slopes and intercepts.

Assuming that transmission was mainly heterosexual, we performed a cross-gendered analysis and estimated the relationship between men's population-level viremia and women's HIV incidence, and between women's population-level viremia and men's HIV incidence. This cross-gendered analysis excluded Ya Tsie, where incidence estimates were not available by sex.

We further evaluated the relationship between prevalence of non-suppression among PLHIV and HIV incidence, for two reasons. First, the relationship between viremia and incidence, even after adjustment for trial, is likely confounded; communities with high prevalence (and thus high population- level viremia) are also likely to have high incidence (Fig 1) due to consistency over time in drivers of HIV transmission, such as mobility and other risk factors. Analyses that fail to account for this will overestimate the effect of decreasing population-level viremia on HIV incidence. Second, non-suppression among PLHIV arguably provides a more policy-relevant target for intervention. We sought to understand how counterfactual HIV incidence would change under a hypothetical intervention to reduce the prevalence of non-suppression among PLHIV, holding HIV prevalence fixed at its observed levels. Specifically, we

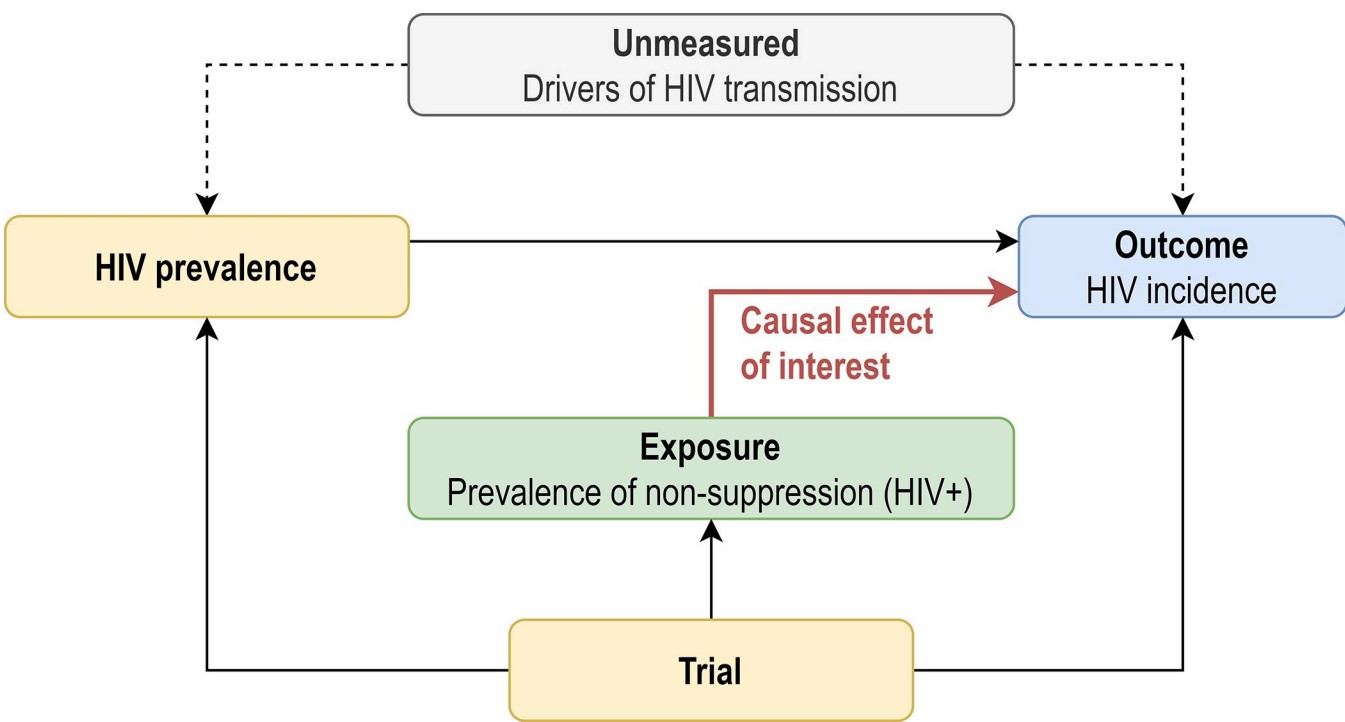

**Fig 1. Simplified causal diagram (directed acyclic graph) for the effect of prevalence of non-suppression among PLHIV on HIV incidence.** In this causal diagram, the effect of prevalence, and thus of population-level viremia on HIV incidence is confounded. In this graph, the effect of non-suppression on HIV incidence is not confounded beyond trial, but adjustment for prevalence, a strong predictor of incidence, is expected to improve precision of estimates.

quantified the relationship between prevalence of non-suppression among PLHIV and HIV incidence using parametric g-computation [19] to estimate parameters of a marginal structural model, implemented using a three-step approach:

1. we fitted a linear regression, using data aggregated at the community level, to model HIV incidence as a function of population-level viremia (at study midpoint), HIV prevalence, and trial; this regression included trial-specific intercepts and interaction terms capturing trial-specific association between HIV prevalence and HIV incidence, which allowed for trial-specific confounding patterns;

2. using the linear regression model from Step 1, we predicted HIV incidence for each community, holding constant the community's trial and observed HIV prevalence, but varying the level of non-suppression among PLHIV from 5% to 65% (in steps of 3%), and thus the level of population-level viremia (viremia is the product of the prevalence of HIV and the prevalence of non-suppression among PLHIV), this extrapolation being within the range in the prevalence of non-suppression that was observed across the 105 study communities (based on the minimum and maximum values given in the S2 Table);

3. we regressed the predicted community-level HIV incidence (across all the extrapolations explored in step 2) on the hypothetical values of the prevalence of non-suppression among PLHIV, allowing for trial-specific intercepts and slopes.

Step 2 can be conceptualized as predicting the counterfactual HIV incidence that each community would have under hypothetical changes in the prevalence of non-suppression among

PLHIV, and step 3 summarises these changes with a "simple" linear regression model of HIV incidence on the prevalence of non-suppression among PLHIV (without needing to also include HIV prevalence as a predictor). We also reported, using step 3 model, that would be the expected incidence per trial if UNAIDS 95-95-95 objectives were reached.

Under the causal assumptions in Fig 1 (including time-ordering) and if the regression models are correctly specified, this approach estimates, for each trial, the average causal effect of a one-unit change in the prevalence of non-suppression among PLHIV on HIV incidence. As these assumptions are unlikely to hold (particularly given the complex time-dependence between prevalence, non-suppression and incidence), this analysis is best interpreted as a summary of the relationship between the prevalence of non-suppression in PLHIV and HIV incidence.

Finally, to illustrate the predicted impact of changes in the prevalence of non-suppression among PLHIV on HIV incidence, for each study and trial arm, we used the model from Step 3 to predict HIV incidence as a function of the trial-arm-specific observed values (averaged across the communities in the same trial arm) of the prevalence of non-suppression among PLHIV at baseline and endline. We then estimated the expected HIV incidence reduction associated with the observed reduction in the prevalence of non-suppression among PLHIV for each trial arm of each study. Specifically, for each trial-arm in each study, we calculated the predicted change in HIV incidence from baseline to endline, summarising this change both as a difference and as a ratio. For both analyses, a bootstrap approach, in which each cluster was sampled with replacement, was applied to compute 95% confidence intervals and p-values [19].

All analyses were performed using R version 4.3.0. A dedicated dataset and an R script are provided for replication (S1 File).

## Results

### Sample characteristics

The four trials were implemented in different epidemiological contexts: baseline HIV prevalence varied from 10% to 31% and baseline prevalence of non-suppression among PLHIV from 25% to 77% across the trials (Table 1). Trials also differed in the population size of communities randomized ($\approx$44 000 people per community in PopART, $\approx$10 500 in SEARCH, $\approx$1 300 aged $\geq$16 in TasP, and $\approx$5 800 in Ya Tsie), location (urban and peri-urban for PopART, rural for SEARCH and TasP, rural and peri-urban for Ya Tsie) and median age of the adult research study population in which the outcomes have been measured (27 years [interquartile range: 22 to 33] for PopART, 29 [20 to 43] for SEARCH, 32 [22 to 52] for TasP, and 40 [33 to 48] for Ya Tsie)[17]. HIV incidence during trial follow-up also varied widely between trials and arms, from 0.25 to 2.27 per 100 person-years.

Communities were also heterogenous (S2 Table). Across all trials, community-level HIV prevalence, measured in 257 929 persons, varied from 2.2% to 41.1%; community-level prevalence of non-suppression among PLHIV, measured in 31 377 persons, from 3.0% to 70.4%; community-population-level viremia, derived from HIV prevalence and non-suppression, from 0.6% to 25.2%; and community-level HIV incidence, measured over 345 844 person-years, from 0.03 to 3.46 per 100 person-years.

### Relationship between population-level viremia and HIV incidence

A strong positive and significant linear relationship was observed between population-level viremia and HIV incidence (Fig 2 and Table 3). The magnitude of this relationship (slopes of the models) was similar for SEARCH, TasP and Ya Tsie: each absolute 10 percentage points

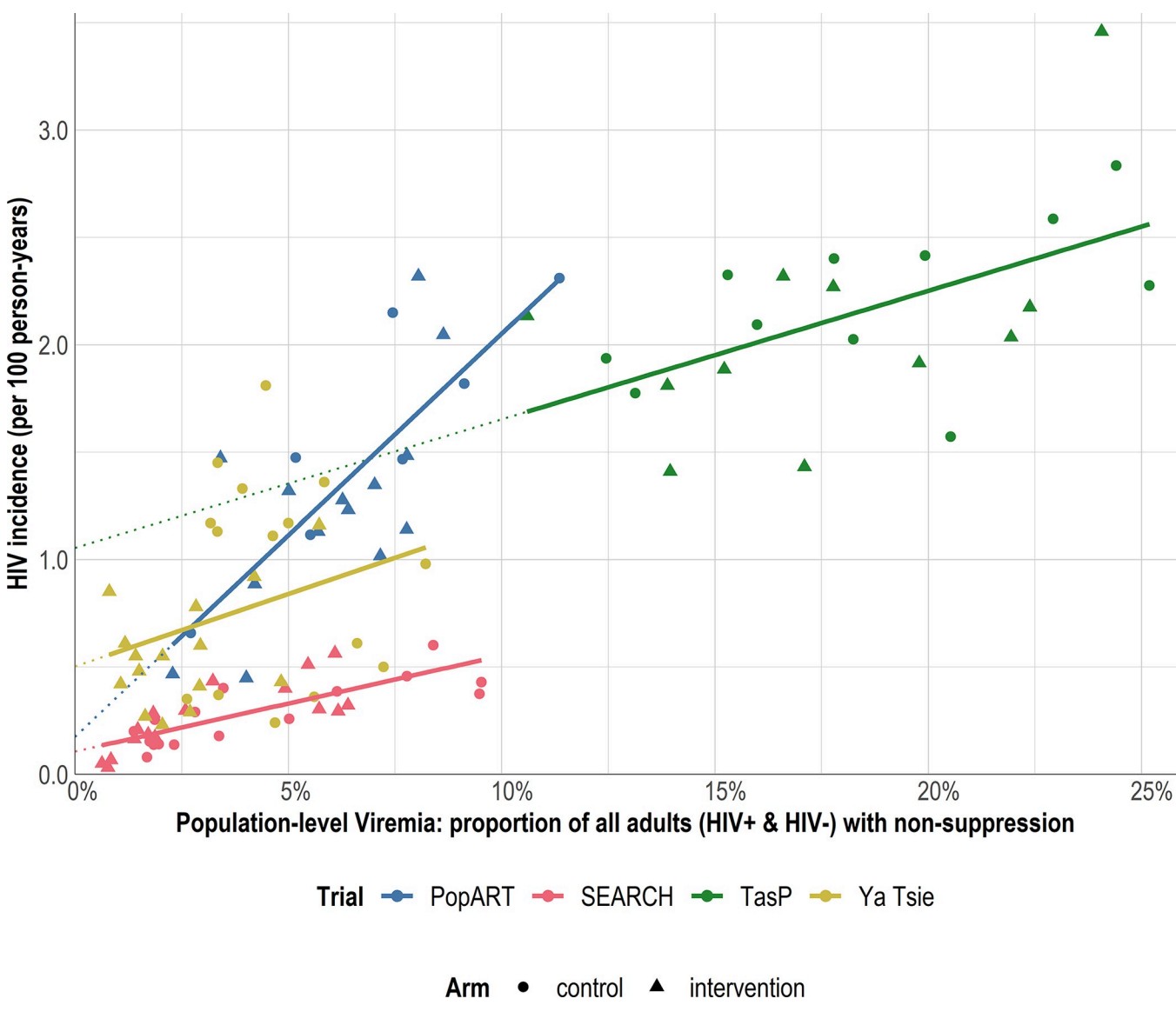

**Fig 2. Relationship at community-level between population-level viremia (the proportion of all adults in the community, both HIV+ and HIV-, with non-suppressed plasma HIV RNA level) and HIV incidence per 100 person-years, by trial.** Each marker represents a community. Lines are based on community-level linear regression.

decrease in population-level viremia was associated with an absolute reduction in expected HIV incidence of 0.446 [95% confidence interval: 0.004 to 0.889] per 100 person-years in SEARCH (p = 0.048), 0.599 [0.258 to 0.939] in TasP (p<0.001) and 0.675 [0.042 to 1.308] in Ya Tsie (p = 0.037). The magnitude was stronger for PopART: 1.877 [1.232 to 2.522] per 100 person-years (p<0.001). Sensitivity analyses further stratified by both trial and arm (S1 Fig and S3 Table) or both trial and country (S2 Fig and S4 Table) yielded similar estimates; all estimated slopes remained positive, except forthe control arm in Ya Tsie (-0.370, -1.434 to 0.694).

Similar results were observed for the cross-gendered analysis (Fig 3 and Table 3): across trials, a decrease in men's population-level viremia was associated with a decrease in women's HIV incidence and a decrease in women's population-level viremia was associated with a decrease in men's HIV incidence. In all trials, the magnitude of the relationship between

**Table 3. Relationship between observed population-level viremia and HIV incidence, by trial and by sex.** Estimates based on community-level linear regressions. Ya Tsie was excluded from the cross-gendered analysis as HIV incidence estimates were not available by sex.

| | Overall (males & females) | | | Men's viremia & Women's incidence | | | Women's viremia & Men's incidence | | |
|---|---|---|---|---|---|---|---|---|---|
| | Coefficient | [95% CI] | p | Coefficient | [95% CI] | p | Coefficient | [95% CI] | p |
| **Slope** (absolute change in expected HIV incidence per 100 person-years per 10 percentage points absolute change in population-level viremia) | | | | | | | | | |
| • PopART | 1.877 | [1.232, 2.522] | <0.001 | 1.812 | [0.643, 2.980] | 0.003 | 0.719 | [0.059, 1.378] | 0.033 |
| • SEARCH | 0.446 | [0.004, 0.889] | 0.048 | 0.509 | [-0.316, 1.335] | 0.223 | 0.393 | [-0.124, 0.910] | 0.134 |
| • TasP | 0.599 | [0.258, 0.939] | <0.001 | 0.476 | [-0.009, 0.961] | 0.054 | 0.115 | [-0.320, 0.550] | 0.600 |
| • Ya Tsie | 0.675 | [0.042, 1.308] | 0.037 | | | | | | |
| **Intercept** (expected HIV incidence per 100 person-years extrapolated to scenario with 0% population-level viremia) | | | | | | | | | |
| • PopART | 0.18 | [-0.26, 0.61] | 0.423 | 1.00 | [0.37, 1.63] | 0.002 | 0.31 | [-0.22, 0.83] | 0.247 |
| • SEARCH | 0.11 | [-0.10, 0.31] | 0.297 | 0.10 | [-0.24, 0.44] | 0.564 | 0.12 | [-0.14, 0.37] | 0.375 |
| • TasP | 1.05 | [0.42, 1.69] | 0.001 | 2.06 | [1.35, 2.77] | <0.001 | 0.63 | [-0.27, 1.53] | 0.165 |
| • Ya Tsie | 0.50 | [0.24, 0.76] | <0.001 | | | | | | |

men's viremia and women's incidence was greater than the relationship between women's viremia and men's incidence: absolute reduction (slope) of 1.812 [0.643 to 2.980] vs 0.719 [0.059 to 1.378] per 100 person-years for each absolute 10 percentage points decrease in

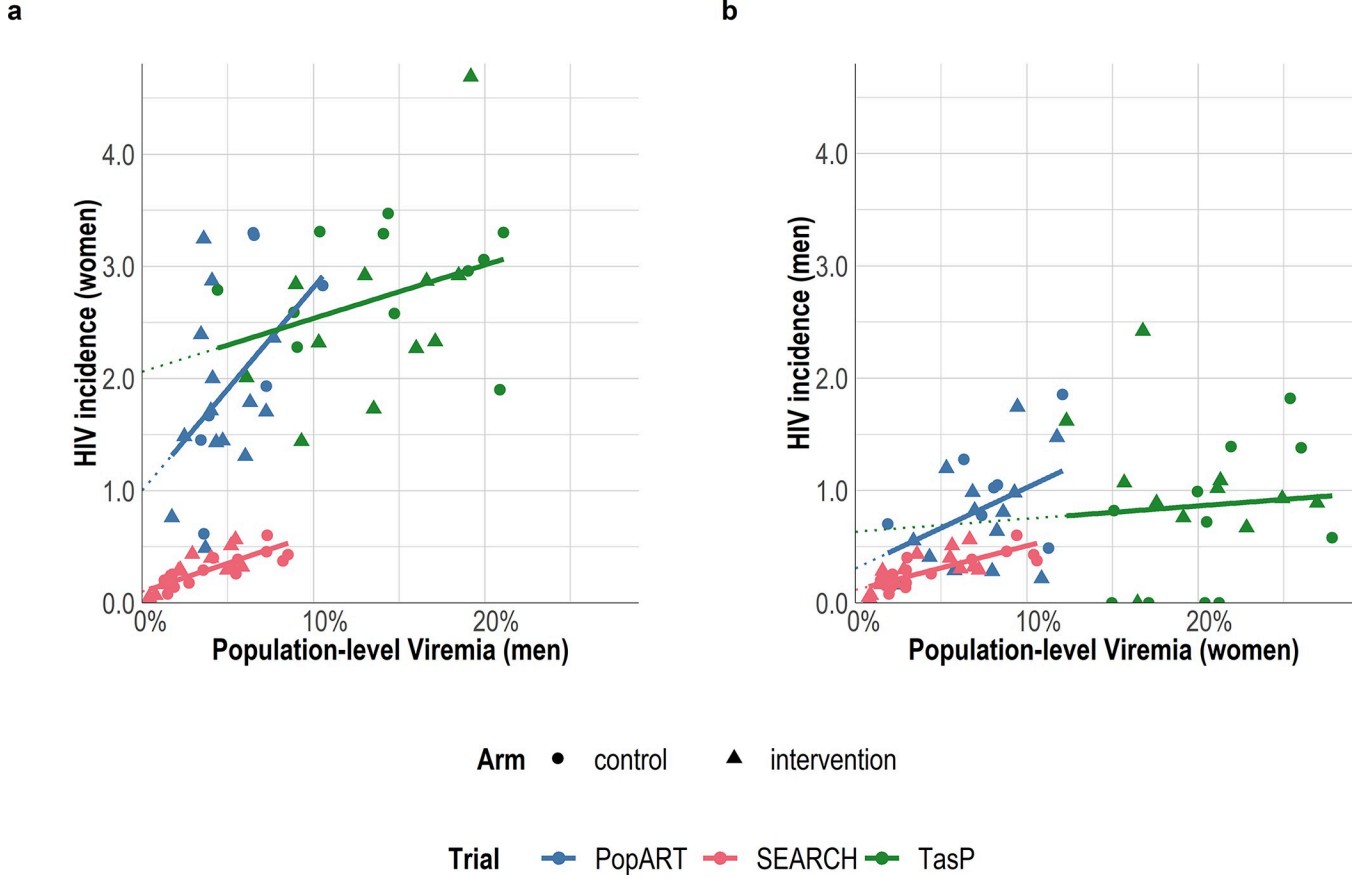

**Fig 3. Cross-gendered relationship at community-level between population-level viremia (the proportion of all adults in the community, both HIV+ and HIV-, with detectable HIV RNA), and HIV incidence per 100 person-years, by trial.** Each marker represents a community. Lines are based on community-level linear regression. Ya Tsie was excluded from this figure as HIV incidence estimates were not available by sex.

population-level viremia in PopART; 0.509 [-0.316 to 1.335] vs 0.393 [-0.124 to 0.910] in SEARCH; and 0.476 [-0.009 to 0.961] vs 0.115 [-0.320 to 0.550] in TasP.

## Relationship between prevalence of non-suppression and HIV incidence

In marginal structural model analysis, a significant relationship between the prevalence of non-suppression among PLHIV and expected counterfactual HIV incidence was also observed (Fig 4 and Table 4). The magnitude (slope) of this relationship was largely consistent across trials; an absolute decrease of 10 percentage points in non-suppression among PLHIV was associated with a decrease in HIV incidence of 0.117 [95% confidence interval: 0.020 to 0.241] per 100 person-years in PopART (p = 0.032), 0.056 [0.009 to 0.114] for SEARCH (p = 0.031), 0.170 [0.028 to 0.348] for TasP (p = 0.033), and 0.158 [0.025 to 0.327] for Ya Tsie (p = 0.033).

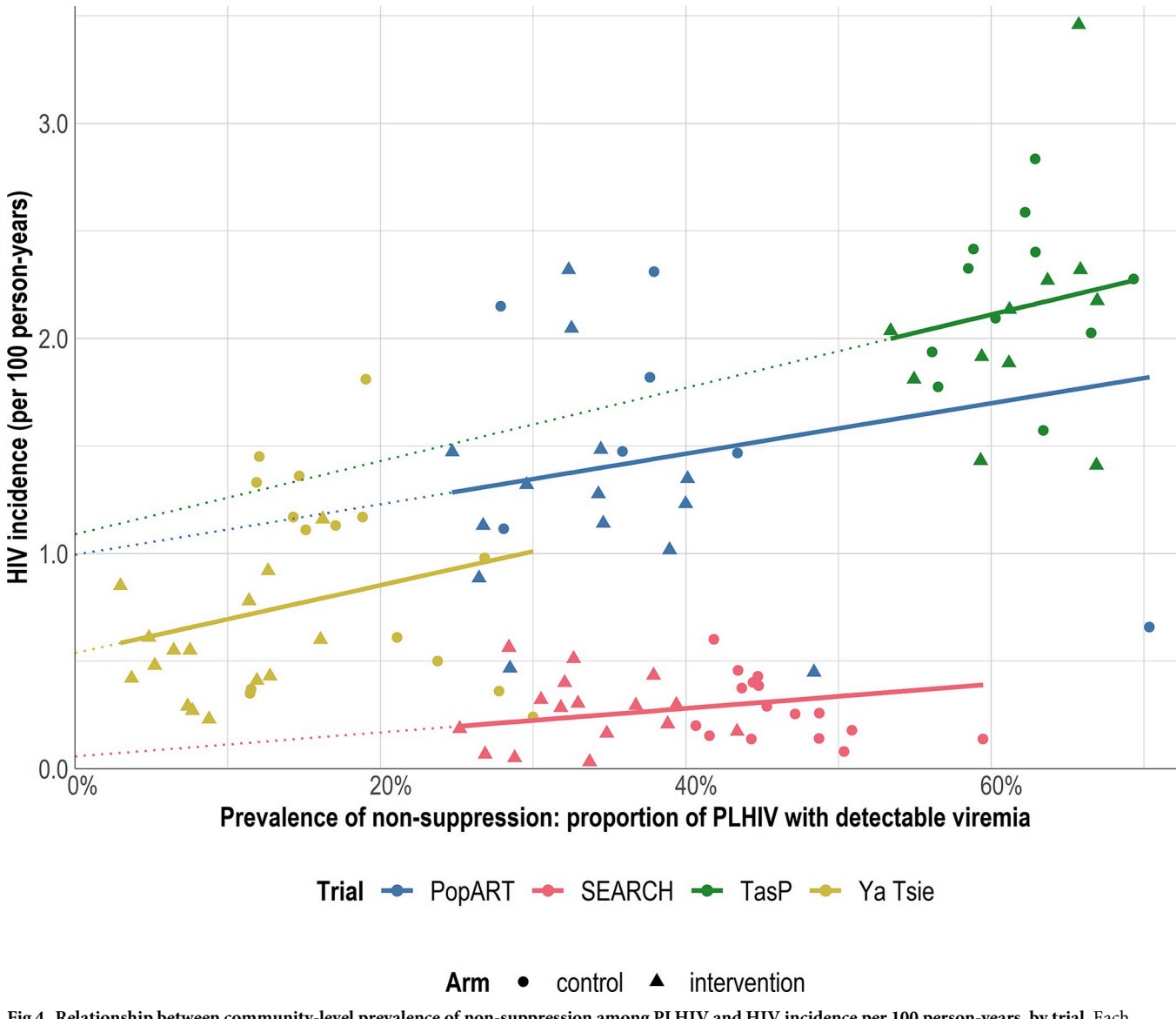

**Fig 4. Relationship between community-level prevalence of non-suppression among PLHIV and HIV incidence per 100 person-years, by trial.** Each marker represents a community. Lines are based on parametric g-computation to estimate the parameters of a marginal structural model, with adjustment for prevalence in the initial regression step. Under assumptions, the slope of the line reflects the change in expected counterfactual HIV incidence across all communities in a trial per hypothetical unit change in prevalence of non-suppression.

**Table 4. Relationship between the prevalence of non-suppression (among PLHIV) and HIV incidence, by trial.** Based on parametric g-computation to estimate the parameters of a linear marginal structural model, with adjustment for prevalence in the initial regression step.

| | Coefficient | [95% CI] | p |
|---|---|---|---|
| **Slope** (under assumptions, absolute change in expected counterfactual HIV incidence per 100 person-years per hypothetical 10 percentage points absolute change in prevalence of non-suppression) | | | |
| • PopART | 0.117 | [0.020, 0.241] | 0.032 |
| • SEARCH | 0.056 | [0.009, 0.114] | 0.031 |
| • TasP | 0.170 | [0.028, 0.348] | 0.033 |
| • Ya Tsie | 0.158 | [0.025, 0.327] | 0.033 |
| **Intercept** (under assumptions, expected counterfactual HIV incidence per 100 person-years extrapolated to scenario with 0% prevalence of non-suppression) | | | |
| • PopART | 0.99 | [0.58, 1.34] | <0.001 |
| • SEARCH | 0.06 | [-0.17, 0.24] | 0.574 |
| • TasP | 1.09 | [0.03, 1.96] | 0.022 |
| • Ya Tsie | 0.54 | [0.34, 0.74] | <0.001 |
| **Expected incidence if UNAIDS 95-95-95 objectives are reached** (under assumptions, expected counterfactual HIV incidence per 100 person-years extrapolated to scenario with 14.2625% prevalence of non-suppression) | | | |
| • PopART | 1.16 | [0.89, 1.40] | |
| • SEARCH | 0.14 | [-0.01, 0.25] | |
| • TasP | 1.33 | [0.52, 2.00] | |
| • Ya Tsie | 0.76 | [0.63, 0.93] | |

Extrapolating to a hypothetical scenario in which 100% of PLHIV were suppressed (a level of non-suppression not present in the observed data) resulted in an estimated HIV incidence (intercept of the model) of 0.99 [95% confidence interval: 0.58 to 1.34] per 100 person-years in PopART, significantly different from zero (p<0.001), 0.06 [-0.17 to 0.24] in SEARCH (p = 0.6), 1.09 [0.03 to 1.96] in TasP (p = 0.022), and 0.54 [0.34 to 0.74] in Ya Tsie (p<0.001).

Extrapolating to scenario where UNAIDS 95-95-95 objectives were reached (corresponding to a prevalence of non-suppression equal to 14.3%) resulted in an estimated HIV incidence of 1.16 [0.89 to 1.40] per 100 person-years in PopART, 0.14 [-0.01 to 0.25] in SEARCH, 1.33 [0.52 to 2.00] in TasP, and 0.76 [0.63 to 0.93] in Ya Tsie.

Across all the trial arms, the prevalence of non-suppression among PLHIV decreased from baseline to the end of the trial (Table 5), with absolute decreases ranging from 9 to 37 percentage points. Based on the relationship estimated between non-suppression among PLHIV and HIV incidence, these observed reductions would be expected to result in absolute reductions in HIV incidence during the trial ranging from 0.11 to 0.39 per 100 person-years, corresponding to a relative reduction ranging from 6.7% to 54.3%.

## Discussion

We evaluated the relationship between HIV incidence and both population-level viremia and non-suppression among PLHIV in four large UTT trials. We found that decreases in population-level viremia were strongly associated with decreased HIV incidence (from 0.446 to 1.877 per 100 person-years absolute decrease in HIV incidence per 10 percentage points absolute decrease in population-level viremia) across a wide range of epidemic settings in eastern and southern Africa. These findings are consistent with prior findings that population-level viremia (which takes into account both the prevalence of non-suppression and HIV prevalence) is a strong predictor of HIV incidence [20], and the importance of incorporating metrics that take into account non-suppression among PLHIV when predicting incidence, especially since

**Table 5. Observed evolution of the prevalence of non-suppression (among PLHIV) between baseline and endline and expected incidence reduction associated with this evolution, per trial.**

| Trial | PopART | | | SEARCH | | TasP | | Ya Tsie | |
|---|---|---|---|---|---|---|---|---|---|
| Country | South Africa & Zambia | | | Kenya & Uganda | | South Africa | | Botswana | |
| Timeline | 2013–2018 | | | 2013–2017 | | 2012–2016 | | 2013–2018 | |
| Arm | C | I (arm A) | I (arm B) | C | I | C | I | C | I |
| **Prevalence of non-suppression (PLHIV)** | | | | | | | | | |
| • at baseline | 49% | 46% | 45% | 59% | 58% | 74% | 77% | 28% | 30% |
| • at endline | 40% | 31% | 31% | 32% | 21% | 55% | 54% | 17% | 12% |
| • absolute decrease (in percentage points) | -9 | -15 | -14 | -27 | -37 | -19 | -23 | -11 | -18 |
| **Expected incidence reduction associated with the observed reduction of non-suppression** | | | | | | | | | |
| • absolute reduction per 100 person-years [95% CI] | -0.11 [-0.22, -0.02] | -0.18 [-0.36, -0.03] | -0.16 [-0.34, -0.03] | -0.15 [-0.31, -0.02] | -0.21 [-0.42, -0.03] | -0.32 [-0.66, -0.05] | -0.39 [-0.80, -0.06] | -0.17 [-0.36, -0.03] | -0.28 [-0.59, -0.05] |
| • relative reduction (%) [95% CI] | 6.7% [1.3, 12.2] | 11.5% [2.2, 21.2] | 10.8% [2.1, 20.1] | 39.0% [8.5, 61.9] | 54.3% [11.7, 86.9] | 13.8% [2.5, 25.3] | 16.3% [3.1, 29.5] | 17.7% [3.8, 27.8] | 28.1% [6.1, 43.2] |

C: Control–I: Intervention–CI: Confidence interval.

the scale-up of antiretroviral treatment. We also build on prior work [20,21] demonstrating a clear association between HIV incidence and prevalence of non-suppression among PLHIV, a community-level metric that can be intervened on more directly using UTT strategies [12,22]. Results were robust to sensitivity analyses, except for the control arm in Ya Tsie, in which a non-significant negative relationship was observed; this may have been due to imprecision resulting from the small number of clusters; the smaller number of observations available to estimate incidence and viremia in this trial may also have contributed.

A cross-gendered analysis, motivated by the assumption that HIV incidence was mainly driven by heterosexual transmission, showed similar results, as also observed in rural Kwa-Zulu-Natal, South Africa [20]. However, the magnitude of the association between women's viremia and men's incidence was lower than that between men's viremia and women's incidence. This finding could reflect that the probability of HIV transmission from women to men is lower than from men to women [23], resulting in lower male incidence than female incidence and, therefore, lower slopes. It is also possible that men's incidence was driven to a greater extent by HIV infections acquired outside the community. Interestingly, observational analysis in the Rakai region of eastern Uganda found larger declines in men's versus women's HIV incidence in the context of increasing HIV viral suppression [21].

Extrapolation of our results to a hypothetical scenario in which 100% of PLHIV were suppressed predicted that residual HIV infections would still occur. As suggested by phylogenetic analysis [24], some HIV seroconversions are likely driven by population mobility and HIV acquisition from outside communities. This finding suggests that improving ART coverage and viral suppression in isolated communities, as occurred in the UTT trial designs, might not be sufficient to stop HIV transmission; challenges in both the time-ordering and estimation of population-level measures may also have contributed. Importantly however, the potential for external infections to drive ongoing HIV transmission in the communities included in this analysis is to a large extent an artifact of the cluster randomized study design; deployment of a UTT strategy at scale would be expected to mitigate this residual source of infections.

The primary outcome of all four trials was HIV incidence and all offered immediate ART to all HIV-positive persons. The trials differed in their approach to testing (Table 1), and thus in the extent to which population-level HIV viremia differed between trial arms. In the two trials (PopART [13] and Ya Tsie [14]) where universal testing was implemented in the intervention arm only, HIV incidence rate was significantly lower in the intervention arm compared to the control arm, while in the two trials where universal testing was provided in both arms (SEARCH [15] and TasP [16]), no significant difference in HIV incidence between arms was observed. In the SEARCH intervention arm HIV incidence was reduced by 32% between year 1 and year 3 [15]. Prior work has suggested that population-level testing strategies implemented in the control arms of the SEARCH and TasP trials reduced the differential in non-suppression, and thus in incidence, observed between arms [12,25,26]. The additional analysis presented here builds on this work by presenting additional estimates of the expected HIV incidence reduction between the beginning and the end of each trial that would be expected in each trial arm given the reduction in the prevalence of non-suppression among PLHIV across the four trials (Table 5).

In Ya Tsie, the prevalence of non-suppression decreased in both arms, but the reduction was much higher in the intervention arm (-18 vs -11 percentage points), as universal testing was not implemented in the control arm. Therefore, the significant difference in overall HIV incidence between arms could partly be explained by the higher expected HIV incidence reduction in the intervention arm associated with better viral control.

In TasP, where repeated testing campaigns were implemented in both arms, the reduction of non-suppression was almost similar between arms (-19 vs -23 percentage points), suggesting that the reduction of HIV incidence due to lower prevalence of non-suppression was also similar between arms. TasP was not able to show a significant difference between arms in terms of cumulative incidence rate.

In SEARCH, a universal testing campaign was implemented at baseline in the control as well as intervention arm and was associated with high linkage to treatment, leading to a substantial reduction of the prevalence of non-suppression in the control arm over time. The difference in HIV incidence between arms was not statistically significant; however, mathematical modelling suggests that the difference would have been substantially larger in the absence of this testing campaign in the control arm [25]. In addition, the trial showed that annual HIV incidence in the intervention arm during the trial decreased by 32% [95% confidence interval: 16% to 44%] [15], from 0.43 cases per 100 person-years between years 0 and 1 to 0.31 between years 2 and 3. In the current analysis, the estimated expected HIV incidence reduction associated with reduction of the prevalence of non-suppression between years 0 and 3 in the intervention arm was 54.3% [11.7% to 86.9%]. The difference may be attributable to differences in the time periods evaluated (given that pre-intervention HIV incidence was not measured), as well as substantial imprecision in both estimates. However, it may also suggest that additional factors such as mobility limited the reduction of HIV incidence.

PopART results also suggest the influence of other drivers of HIV incidence, beyond non-suppression. While the reduction in the prevalence of non-suppression was similar in the two intervention arms A & B (-15 and -14 percentage points respectively), the cumulative HIV incidence rate was different in both arms (1.45 cases per 100 person-years in arm A vs 1.06 in arm B) [13]. Consequently, HIV incidence was significantly lower in arm B vs the control arm, while the difference between arm A and the control arm was not statistically significant. As a similar reduction in HIV incidence associated with the reduction in the prevalence of non-suppression was expected in arms A and B, it suggests that other factors may have played a role in the evolution of HIV incidence in the two arms.

Our analysis is subject to limitations. There are likely uncontrolled confounding factors of the relationship observed between HIV incidence and both population-level viremia and the prevalence of non-suppression among PLHIV. The viremia-incidence relationship, in particular, will be confounded by any shared factors that vary among communities within a trial and drive both HIV prevalence and HIV incidence. While shared factors affecting non-suppression among PLHIV and incidence may be less of a concern, particularly after adjustment for trial, some degree of residual confounding remains likely. Interpretation of the observed associations is further complicated by the lack of clear time-ordering between exposures and outcomes; non-suppression among PLHIV during the study may have affected HIV incidence, while incident HIV infections in turn may have contributed to non-suppression; our results thus provide a summary of the association between these two complex time-dependent processes. Finally, the diversity of data collection tools, methods, and timing of measures is an additional limitation of our analysis, complicating comparison of estimates across trials.

Interestingly, the estimated relationship between the prevalence of non-suppression among PLHIV and HIV incidence was surprisingly consistent across trials, suggesting that one major contributor to the heterogeneity in the observed association between population level viremia and incidence (i.e. apparent "effect modification") across trials may have been the presence of different confounding patterns (i.e., differences in the extent of unmeasured shared determinants of HIV prevalence and incidence) across the studies. Nonetheless, the effect of local population-level viremia on HIV incidence is likely to vary across settings, as a result of differences in factors such as including mobility, sexual network characteristics and risk behaviours, and, particularly going forward, coverage of biomedical prevention such as pre-exposure prophylaxis.

In summary, observational analysis of pooled data from the four UTT trials supports the utility of population-level viremia as a predictor of incidence, and thus as a tool to allow for the appropriate targeting of prevention interventions; however, generating accurate population level estimates requires care to account for non-representative participation [27,28]. It further supports the ability of UTT approaches to impact HIV incidence by reducing the prevalence of non-suppression among PLHIV. The magnitude of HIV incidence reduction with UTT approaches will differ in different contexts due to structural drivers of incidence. Policies will be more effective if they are consistently applied at a larger geographical scale due to population mobility and external HIV acquisition, and if implementation is adapted to local context. Based on the joint experience of the UTT trials, implementation of universal HIV testing approaches, coupled with interventions to leverage linkage to treatment, can reduce HIV acquisition at population level.

## Supporting information

**S1 Fig. Relationship at community-level between population-level viremia (the proportion of all adults in the community, both HIV+ and HIV-, with non-suppressed plasma HIV RNA level) and HIV incidence per 100 person-years, by trial and trial arm.** Each marker represents a community. Lines are based on community-level linear regression.
(TIFF)

**S2 Fig. Relationship at community-level between population-level viremia (the proportion of all adults in the community, both HIV+ and HIV-, with non-suppressed plasma HIV RNA level) and HIV incidence per 100 person-years, by trial and country.** Each marker represents a community. Lines are based on community-level linear regression.
(TIFF)

**S1 Table. Additional trial-specific details regarding methodologic approach to estimation of key measures.**
(DOCX)

**S2 Table. Median [minimum–maximum] values of HIV prevalence, prevalence of non-suppression at midpoint, population-level viremia and HIV incidence across communities, per trial.**
(DOCX)

**S3 Table. Relationship between observed population-level viremia and HIV incidence, by trial and trial arm.** Estimates based on community-level linear regressions.
(DOCX)

**S4 Table. Relationship between observed population-level viremia and HIV incidence, by trial and country.** Estimates based on community-level linear regressions.
(DOCX)

**S1 File. Dataset and R script.**
(ZIP)

## Acknowledgments

The Universal Test and Treat Trial Consortium (UT3C) is composed of the teams of five trials: the Ya Tsie BCPP trial (ClinicalTrials.gov NCT01965470), the MaxART trial (ClinicalTrials.gov NCT02909218), the HPNT 071 PopART trial (ClinicalTrials.gov NCT01900977,), the SEARCH trial (ClinicalTrials.gov NCT01864603) and the ANRS 12249 TasP trial (Clinical-Trials.gov NCT01509508).

The authors thank all members of the study teams including the Universal Testing and Treatment Trials Consortium (UT3C), the communities where we work, our sponsors, and all the policy makers, organizations and community members around the world engaging in efforts of HIV epidemic control. The views expressed here represent those of the authors only and do not necessarily represent the official position of the funding agencies.

The TasP trial was done with the support of Merck and Gilead Sciences, which provided the Atripla drug supply. The Africa Health Research Institute (previously Africa Centre for Population Health, University of KwaZulu-Natal, South Africa) receives core funding from the Wellcome Trust, which provides the platform for population-based and clinic-based research at AHRI.

## Author Contributions

**Conceptualization:** Joseph Larmarange, Pamela Bachanas, Timothy Skalland, Laura B. Balzer, Collins Iwuji, Sian Floyd, Lisa A. Mills, Deenan Pillay, Diane Havlir, Moses R. Kamya, Helen Ayles, Kathleen Wirth, François Dabis, Richard Hayes, Maya Petersen.

**Data curation:** Joseph Larmarange, Timothy Skalland, Sian Floyd, Maya Petersen.

**Formal analysis:** Joseph Larmarange, Laura B. Balzer, Maya Petersen.

**Funding acquisition:** Diane Havlir, Moses R. Kamya, Helen Ayles, François Dabis, Richard Hayes, Maya Petersen.

**Investigation:** Collins Iwuji, Deenan Pillay, Diane Havlir, Moses R. Kamya, Helen Ayles, François Dabis, Richard Hayes, Maya Petersen.

**Methodology:** Joseph Larmarange, Laura B. Balzer, Richard Hayes, Maya Petersen.

**Software:** Joseph Larmarange.

**Validation:** Laura B. Balzer.

**Visualization:** Joseph Larmarange.

**Writing – original draft:** Joseph Larmarange, Maya Petersen.

**Writing – review & editing:** Joseph Larmarange, Pamela Bachanas, Timothy Skalland, Laura B. Balzer, Collins Iwuji, Sian Floyd, Lisa A. Mills, Deenan Pillay, Diane Havlir, Moses R. Kamya, Helen Ayles, Kathleen Wirth, François Dabis, Richard Hayes, Maya Petersen.

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
