## [Decision Letter · Decision Letter 0]

27 Mar 2023

PGPH-D-23-00266

Population-level viremia predicts HIV incidence at the community level across the Universal Testing and Treatment Trials in eastern and southern Africa

Dear Dr. Larmarange,

Thank you for submitting your manuscript to PLOS Global Public Health. The reviewers and I read the manuscript with much interest and were impressed, and we invite you to submit a revised version of the manuscript that addresses the points raised during the review process.

The reviewers have provided several comments and suggestions which I would appreciate you to consider in your revisions. In particular, several reviewers noted the apparent different slope of relationship between community viremia and HIV incidence in across the PopART communities compared to the other studies, and whether this changed if the Western Cape and Zambia study sites were considered separately. They also raised questions about the choice of linear functional form and whether others were considered.

We look forward to receiving your revised manuscript.

Kind regards,

Jeffrey William Eaton

Academic Editor

Journal Requirements:

2. We ask that a manuscript source file is provided at Revision. Please upload your manuscript file as a .doc, .docx, .rtf or .tex.

3. Please provide separate figure files in .tif or .eps format only and remove any figures embedded in your manuscript file. Please also ensure that all files are under our size limit of 10MB.

Additional Editor Comments (if provided):

Reviewers' comments:

Reviewer's Responses to Questions

**Comments to the Author**

1. Does this manuscript meet PLOS Global Public Health’s publication criteria? Is the manuscript technically sound, and do the data support the conclusions? The manuscript must describe methodologically and ethically rigorous research with conclusions that are appropriately drawn based on the data presented.

Reviewer #1: Yes

Reviewer #2: Yes

Reviewer #3: Yes

Reviewer #4: Yes

Reviewer #5: Yes

2. Has the statistical analysis been performed appropriately and rigorously?

Reviewer #1: Yes

Reviewer #2: Yes

Reviewer #3: Yes

Reviewer #4: Yes

Reviewer #5: Yes

3. Have the authors made all data underlying the findings in their manuscript fully available (please refer to the Data Availability Statement at the start of the manuscript PDF file)?

Reviewer #1: Yes

Reviewer #2: Yes

Reviewer #3: Yes

Reviewer #4: Yes

Reviewer #5: Yes

4. Is the manuscript presented in an intelligible fashion and written in standard English?

Reviewer #1: Yes

Reviewer #2: Yes

Reviewer #3: Yes

Reviewer #4: Yes

Reviewer #5: Yes

5. Review Comments to the Author

Reviewer #1: This article presents an analysis combining the data from 4 trials of UTT in different countries that came to similar but not entirely identical findings and pools the data to explore the relationship between population viremia and HIV incidence (among other relationships). The methods are strong and most of the limitations are discussed. The results are not at all surprising in terms of the form of the relationship, but with analyses like these the value is in the actual size of the effects. Below are specific comments about the work.

I am strongly supportive of what the authors have done. However, I do think that, as they note, the effects of any UTT approach is going to be population specific and as such, I think there needs to be some more discussion of this issue. There is a nice explanation of the potential for residual confounding but here I’m more concerned about effect modification. What are the factors that are likely to explain different effect sizes. Can the authors speculate in the discussion?

In the methods, I found it a bit hard to follow what time periods each data corresponded to and how many observations each cluster could contribute. Could a cluster contribute one measure of prevalence of suppression or two? And I assume they could only contribute one measure of incidence? If that’s correct, how do the two measures of viremia relate to the one measure of incidence? Of course, if I’ve misunderstood a brief sentence in the methods could clarify that.

For the baseline HIV prevalence and non-suppression numbers, can you provide confidence intervals so we can see how much uncertainty there is in each community?

In reading through the methods, my first thought was, why just a linear function, why not something more curved. Reviewing the data in the figures suggests that maybe a linear model is the right one, but I can’t be sure. Did you try something like a quadratic? If not, it would be worth checking it doesn’t add to the fit of the model. And either way, it is worth explaining in the methods why you chose a linear function.

The results for PopART are so different from the rest, it really needs some explaining. There isn’t really any discussion of this in the discussion section. What explains this much larger effect size in one trial vs another. As Ive noted above, I think there could be lots of explanations that should be explored, but these don’t come up in the discussion.

Given the differences in testing protocols across studies it might help to see an analysis strategies by whether the control arms got universal testing.

It would also be helpful to see the results stratified by intervention and control arm to see if the effect of the intervention changed the relationship.

I don’t know if this is possible, partly because of the timing of the data, but given the different trials found somewhat different results whether you could look at change in the exposures in relation to changes in the outcome. Perhaps change from baseline to midpoint in population viremia and change in incidence from baseline to endpoint?

I could just be misunderstanding, but is Table 1 missing the prevalence of unsuppressed?

In figures 2-4 it would help for each symbol’s size to represent the size of the cluster to give some sense for how precisely each of these values is likely to be measured.

Reviewer #2: Many thanks for the opportunity to review this manuscript.

This is a well written, well conceptualized manuscript that uses novel methods to estimate the magnitude of the impact of changes in viral suppression among people living with HIV (and the community-level viremia) on HIV incidence. Here are some suggestions for authors to consider:

Introduction

• Line 116 – It is not immediately clear what the authors mean by “cross gendered analysis”. I suggest making this explicit in the introduction rather than further down, as it is now.

Methods

• Line 166 - the authors statement of “community-level” linear regression was unclear to me at first. After looking at their R code (thank you for providing it!), they use study-level, aggregate viremia as an independent variable and study as the fixed-effect. If so, this could be better phrased as aggregate (or ecological, study level…) analysis, rather than community (N = 105 clusters/communities, as stated in line 126) level. In short, I would clarify the language by stating that that HIV incidence was regressed on aggregate study-level viremia, or something along these lines.

• Line 173 – in the analysis for the association between prevalence of non-suppression among PLHIV and HIV incidence, prevalence is measured at year two or endline in 3 out of 4 trials. Meanwhile incidence is measured at different points in time between 0-36 months. Therefore temporality between the two, especially given the authors’ use of causal methods (and sometimes language) is questionable. The authors briefly address this issue with one sentence in limitations, but I think it deserves further discussion/acknowledgement throughout the methods and discussion.

• Line 204 and 205 – the authors refer to causal assumptions in the DAG and to their estimate as the “average causal effect of …”. I would be careful using this and other causal language, even though they are using a G-methods to conduct the analysis for the following reasons, which I should be addressed in the paper (discussion or limitations):

o The estimates are calculated at an aggregate/ecological/study level not individual, which is a barrier to making any causal statements.

o One of identifiability conditions for causality is consistency (the observed outcome for every treated individual equals her outcome if she had received treatment) and requires a well-defined, consistent intervention. As the authors pointed out throughout the paper the trial interventions varied widely, further questioning the interpretation of estimates as causal.

o If the HIV incidence in the community is high, we are also more likely to see more unsuppressed individuals in the community (higher viremia), due to the time it takes for linkage to care to take place further questioning the directionality of the estimates.

• Line 208 – authors should clarify “community-based bootstrap” do they mean that “study” is their resampling unit when doing the bootstrap analysis?

• Line 214 – I suggest using “associated with” or “predictive of” instead of “due to”.

Results

• Figure 1 – As the authors present it, the adjustment of the Viremia  HIV incidence relationship for HIV prevalence (the collider) opens non-causal paths between E and O. Would authors consider adjustment for some of the “beyond-trial” drivers of HIV acquisition e.g. HIV prevalence outside of the trial? This is especially relevant since external HIV acquisition could still occur outside of the community , given the cluster randomized nature of the studies.

• Table 3 and 4 – minor point, but some values are rounded at 2 while others at 3 decimal points. Best to stick with one for consistency.

• Thank you for providing your code and data for replication – always helpful for students!

Discussion

• Line 303 – Authors hypothesize that HIV infection in men could be driven by infections acquired outside of the community (again in 308/309). Further, trials are often at risk of selection bias, such that people (communities in this case) enrolled are not representative of the HIV transmission patterns in the general population. Given this, have the authors considered adjusting for HIV prevalence outside of the trial community to account for any residual confounding due to infections acquired outside of the trial “boundaries?”

• Line 307 – this is a minor point, but have the authors considered reporting on the 95% VLS scenario instead of 100% to better align with the UNAIDS goals?

Reviewer #3: This study by Joseph Larmarange and colleagues describes the relationship between the prevalence of unsuppressed HIV viraemia and HIV incidence, as well as the relationship between the proportion of PLHIV who are unsuppressed and HIV incidence, at a population level. As might be expected, and in line with previous studies, both relationships are strongly positive, although it is interesting to see heterogeneity in the extent of the positive relationship. The study is based on data from four cluster-randomized controlled trials that evaluated the effectiveness of ART as a population-level intervention to reduce HIV incidence (“treatment as prevention”). This is a large and important dataset, which adds considerable weight to the argument that reductions in HIV viraemia can be expected to reduce HIV incidence – despite the four trials being somewhat inconsistent in their findings about the impact of treatment as prevention. The study is well-written and clear.

Minor comments

1. The assumption of a linear relationship between HIV incidence and the level of viraemia could be questioned, particularly in the presence of significant heterogeneity in risk. (Key populations such as PWID, FSW and MSM might contribute disproportionately to transmission even though they only account for a small fraction of the population.) Instead of fitting a model of the form y = a + bx, where y is the incidence rate and x is the prevalence of unsuppressed viraemia, one might fit a model of the form y = a + x^b, or even just y = x^b. The authors do a good job of explaining why it’s appropriate to include the intercept term (a) in the context of a cRCT, but taken out of context (for example in a mathematical model of the impact of treatment as prevention at a national level), it could be troublesome. I suspect that fitting these alternative regression models would do little to change the overall conclusions of the paper, and I don’t feel strongly that they should be included, but I include this point as an observation, which the authors can ignore if they choose.

2. Abstract, line 122: “increase in suppression” rather than “reduction of non-suppression”.

3. Table 1: It’s not clear what the (12-15) in the row headings at the bottom represents. Maybe these are citations, though it’s not clear they’re necessary.

4. In Figure 2, PopART looks like an outlier, in terms of the steepness of the slope. What would explain this? Maybe there is heterogeneity between Western Cape and Zambia? Did the authors consider fitting separate lines for the Western Cape and Zambian sites?

5. Last paragraph of p. 14: Another explanation for the difference between male and female slopes (which is slightly different from the point about differences in transmission probabilities) is that female incidence is higher than male incidence, so you’d expect the slope to be greater (in absolute terms).

6. In the same paragraph, (line 304) “also found” suggests the results are consistent, but they actually seem to be inconsistent (?). If the Rakai study found greater male reductions in HIV incidence, would that not suggest a greater slope in men?

7. A general grammatical issue: The authors tend to talk about “an increase of HIV incidence” (or other metrics) when it would be more conventional to say “an increase in HIV incidence”. Similarly for “a reduction of…”. One would say “an increase of 5%” but not “an increase in 5%”.

Reviewer #4: I enjoyed reading this paper and found it well conceived and explained. I have only a few minor comments:

Line 248- “HIV prevalence hetergeneous…”

Here I think it would be more helpful to give the ranges across all the trials combined instead of across the whole sample as that would better support the point you are making. eg. the within-trial range in HIV prevalence was between 20 and 30 percentage points etc.

Line 294 “importance of incorporating metrics of non-suppression among PLHIV when predicting incidence,” . Should that be population viremia, since that paragraph is refering to viremia and the following paragraph is about non-suppression.

Line 156- Population-level viremia, also known as the prevalence of detectable virus and defined as the prevalence of non-suppressed HIV infections among all resident adults, irrespective of HIV status, was obtained by multiplying the prevalence of non-suppression (among PLHIV) at midpoint and HIV prevalence.

This definition doesn’t quite match with the data given in the supplementary material i.e. the viremia estimate isn’t the product of prevalence and non-supression but perhaps just a rounding difference but would be worth noting.

Reviewer #5: Thank you for the opportunity to review this comprehensive, informative analysis on population viremia and HIV incidence in the four UTT trials in eastern and southern Africa. I recall seeing this work presented in conference, and I’m very pleased to finally see it summarized for publication. The work is novel, thorough, and provides critical epidemiological insights. The manuscript was well written, analysis well done, and the tables and figures are clear and to the point. I have only minor comments.

1) How were biases in study participation dealt with in these analyses? How sensitive would results be to different assumptions about differential participation in the survey by HIV and viremia status? Could selection biases into the incidence and viremia cohorts potentially explain to some extent why achieving 100% suppression in PLHIV would not achieve 100% reductions in HIV incidence?

2) It would be helpful if the authors referred to community level prevalence estimates accordingly throughout the manuscript. There are times when the authors are referring to overall prevalence estimates in the cohorts and then to community level estimates. Even in the abstract I found it confusing.

3) Why were the VL suppression cutoffs chosen for the various cohorts? Were these the lower limit of detection of the assays used?

4) Could the authors please expand briefly upon how community bootstrapping was done?

5) The authors hypothesize that the stronger relationship between male viremia and female HIV incidence vs. female viremia and male HIV incidence could be due to either lower probability of male infection or higher probability of external HIV acquisition among men. Is it possible to adjust for community male circumcision prevalence in these trials, or was prevalence too low? The authors mention external male infection from outside the community, but observations could also be due to unsampled key populations or possibly higher male contact rates. It might be worth expanding here. Is it also possible the weaker correlation with male incidence might be because we are under sampling male HIV incident cases?

6) The author recommend surveillance for population viremia, but don’t specify exactly how that should best be done. What are their recommendations?

6. PLOS authors have the option to publish the peer review history of their article (what does this mean?). If published, this will include your full peer review and any attached files.

**Do you want your identity to be public for this peer review?** For information about this choice, including consent withdrawal, please see our Privacy Policy.

Reviewer #1: No

Reviewer #2: No

Reviewer #3: No

Reviewer #4: No

Reviewer #5: No

---

## [Decision Letter · Decision Letter 1]

10 May 2023

PGPH-D-23-00266R1

Population-level viremia predicts HIV incidence at the community level across the Universal Testing and Treatment Trials in eastern and southern Africa

Dear Dr. Larmarange,

Thank you for submitting your revised manuscript to PLOS Global Public Health. Following review of the revised manuscript, the reviewers and I felt that most of the comments have been comprehensively addressed. Thank you for the revisions and comprehensive responses.

The main point that we would like further considered is further discussion explaining the results around heterogeneity, particularly the important explanation "This suggests that what appeared to be a much larger “effect” in POPART was likely largely driven by differences in confounding of the prevalence/incidence relationship." would be helpful to elaborate in the Discussion to communicate the causal inference results to a wider non-specialist audience (and see also further comments from Reviewer #1).

I tend to agree with the reviewer regarding the interpretation of confidence intervals from a local census.

We look forward to receiving your revised manuscript.

Kind regards,

Jeffrey William Eaton

Academic Editor

Journal Requirements:

Additional Editor Comments (if provided):

Reviewers' comments:

Reviewer's Responses to Questions

**Comments to the Author**

1. If the authors have adequately addressed your comments raised in a previous round of review and you feel that this manuscript is now acceptable for publication, you may indicate that here to bypass the “Comments to the Author” section, enter your conflict of interest statement in the “Confidential to Editor” section, and submit your "Accept" recommendation.

Reviewer #1: (No Response)

Reviewer #2: All comments have been addressed

2. Does this manuscript meet PLOS Global Public Health’s publication criteria? Is the manuscript technically sound, and do the data support the conclusions? The manuscript must describe methodologically and ethically rigorous research with conclusions that are appropriately drawn based on the data presented.

Reviewer #1: Yes

Reviewer #2: Yes

3. Has the statistical analysis been performed appropriately and rigorously?

Reviewer #1: Yes

Reviewer #2: Yes

4. Have the authors made all data underlying the findings in their manuscript fully available (please refer to the Data Availability Statement at the start of the manuscript PDF file)?

Reviewer #1: Yes

Reviewer #2: Yes

5. Is the manuscript presented in an intelligible fashion and written in standard English?

Reviewer #1: Yes

Reviewer #2: Yes

6. Review Comments to the Author

Reviewer #1: Overall I'm happy with the changes and commend the authors on the revisions, but there are two points that I think are worth coming back to.

First, the issue of heterogeneity. It seems very unlikely that the effects of changes in population level viremia on incidence wouldn't be population specific. As such, the note that much of the effect disappears when accounting for prevalence seems not sufficient. I'd like to see this discussed more, not just listed as a limitation as it doesn't seem like your data is sufficient to answer the question definitively and so more comment on this seems important.

Second, you note that you don't need confidence intervals on the prevalence because this is a population census. I disagree with this approach and take the view expressed in Modern Epidemiology that all populations, even if a census, can be seen as a sample of a larger super population. I wouldn't go to the mat over this and leave it to the editor to decide if this an important enough change, but it does seem worth noting.

Reviewer #2: The authors have appropriately addressed my comments regarding the paper. Thank you taking the time.

7. PLOS authors have the option to publish the peer review history of their article (what does this mean?). If published, this will include your full peer review and any attached files.

**Do you want your identity to be public for this peer review?** For information about this choice, including consent withdrawal, please see our Privacy Policy.

Reviewer #1: **Yes: **Matthew Fox

Reviewer #2: No

---

## [Editor Report · Decision Letter 2]

21 Jun 2023

Population-level viremia predicts HIV incidence at the community level across the Universal Testing and Treatment Trials in eastern and southern Africa

PGPH-D-23-00266R2

Dear Dr. Larmarange,

We are pleased to inform you that your manuscript 'Population-level viremia predicts HIV incidence at the community level across the Universal Testing and Treatment Trials in eastern and southern Africa' has been provisionally accepted for publication in PLOS Global Public Health.

Best regards,

Jeffrey William Eaton

Academic Editor